# The Application of Porous Scaffolds for Cardiovascular Tissues

**DOI:** 10.3390/bioengineering10020236

**Published:** 2023-02-10

**Authors:** Tatsuya Watanabe, Salha Sassi, Anudari Ulziibayar, Rikako Hama, Takahiro Kitsuka, Toshiharu Shinoka

**Affiliations:** 1Center for Regenerative Medicine, The Abigail Wexner Research Institute at Nationwide Children’s Hospital, Columbus, OH 43205, USA; 2Department of Surgery, Nationwide Children’s Hospital, Ohio State University, Columbus, OH 43205, USA; 3Department of Cardiothoracic Surgery, The Heart Center, Nationwide Children’s Hospital, Columbus, OH 43205, USA

**Keywords:** porous scaffold, tissue engineering, cardiovascular, vascular grafts, valves, cardiac patches, regenerative medicines

## Abstract

As the number of arteriosclerotic diseases continues to increase, much improvement is still needed with treatments for cardiovascular diseases. This is mainly due to the limitations of currently existing treatment options, including the limited number of donor organs available or the long-term durability of the artificial organs. Therefore, tissue engineering has attracted significant attention as a tissue regeneration therapy in this area. Porous scaffolds are one of the effective methods for tissue engineering. However, it could be better, and its effectiveness varies depending on the tissue application. This paper will address the challenges presented by various materials and their combinations. We will also describe some of the latest methods for tissue engineering.

## 1. Introduction

Our understanding of medical and surgical treatment [1,2,3,4,5] and postoperative management have progressed steadily year by year [6,7]. Demands for its excellence continue to grow. The American Heart Association (AHA) reports that there are 960,000 new cases of heart failure each year in the United States alone [8]. The annual number of deaths related to heart failure has reached 640,000 [8]. Although heart transplantation is the most effective treatment for end-stage severe heart failure [9,10], the number of patients who can receive such treatment is limited, due to the number of donor organs and institutional and ethical issues. Heart failure significantly reduces the patient’s quality of life, has a high mortality rate, and substantially contributes to increased medical costs [11,12,13,14]. Various diseases cause heart failure, including ischemic heart disease, valvular disease, or heart myopathy.

Atherosclerosis is the leading cause of ischemic heart disease and peripheral artery disease. Ischemic heart disease accounts for 50% of heart failure in the United States [15]. The best way to treat these diseases is to replace the diseased vessels with autologous grafts. However, the number of available autologous grafts is limited, due to the adverse vessel quality caused by the comorbidities these patients often face, such as high blood pressure and diabetes. Therefore, artificial vessel grafts are produced. The currently used artificial blood vessels are mainly made of expanded polytetrafluoroethylene (ePTFE). However, there are problems, such as poor patency, stenosis, and occlusion of artificial vessels compared to autologous vessels [16]. In native vessels, the vascular wall has a three-layered structure consisting of the intima, media, and adventitia. Each layer has a function, such as preventing thrombus formation, contraction, and expansion [17]. However, when a vascular prosthesis is used, it is composed of a single-layer structure. Without a functional intimal layer, thrombus formation becomes a severe problem until it is covered with an autologous intima. Also, artificial grafts do not have dynamic functions, such as constriction and expansion properties. In addition, extensive surgery is required to excide whole artificial grafts when infection occurs, since the infiltration of immune cells does not happen in synthetic blood vessels [18]. In the pediatric field, once a conventional artificial blood vessel is implanted, the size of the artificial blood vessel does not change, contrary to body growth. Therefore, those patients need repeated surgery to replace the graft with a new, larger graft after the patients have grown up [19].

In valvular disease, age-related degeneration causes valvular stenosis and regurgitation accompanied by calcification. The structure of the native valve is a three-layer tissue structure rich in ECM with low cellular components. Anatomically, it has bicuspid or tricuspid cusps to withstand pressure loads [20]. The definitive treatment for valvular disease is to replace the damaged valve with a functioning valve. Mechanical valves and tissue valves are commonly used artificial valves. While mechanical valves are highly durable and can be used for an extended period, they require permanent anticoagulant therapy, which poses serious problems such as bleeding.

On the other hand, bioprosthetic valves, made from animal valves or pericardium, do not require long-term anticoagulation therapy. However, it is reported that 40% of them require retreatment intervention in 20 years [21]. Furthermore, when performing valve replacement surgery due to congenital heart disease, the size of the transplanted valve poses a significant problem. If the patient undergoes surgery at a young age, the size of the implanted valve becomes unsuitable as the body grows, requiring reoperation. Therefore, surgical treatment using autologous pericardium with the expectation of tissue growth has been developed, but its long-term durability is similar to the results of biological valves [22].

One of the most common causes of cardiomyopathy is ischemic cardiomyopathy, but its treatment is complex. Clinically, coronary artery bypass surgery and percutaneous revascularization have been performed. Still, revascularization after necrosis of the myocardium does not regenerate cardiomyocytes [23]. As such, myocardial regeneration remains an important research topic these days. Cardiac tissue consists of cardiomyocytes, fibroblasts, progenitor cells, and conduction systems. Most of the current cardiac regeneration is focused on the regeneration of cardiomyocytes. The biggest problem in myocardial regeneration is that mature myocardium has almost no proliferative capacity [24]. Therefore, many attempts have been made to replace damaged myocardial tissue with cultured cardiomyocytes, such as injecting cultured cells or engineered sheets. However, in these “cell therapies,” the low viability of the cells upon transplantation and leakage from the implanted site are significant problems. Few promising results have been reported in this field [25].

The main objective of regenerative medicine is to apply engineering technologies to create biological products that can regenerate damaged organs or tissues. This field of medicine includes various principles and understanding from molecular biology, material science, and engineering. These backgrounds indicate the need for new therapeutic strategies in cardiovascular surgery. Since its inception, tissue engineering has made it possible to create a source of healthy neoplastic tissue intended to replace, repair, and regenerate diseased or damaged tissues. Many tools have been developed, including generating and identifying new cell sources, controlled biophysical and biochemical stimuli delivery, and synthesizing increasingly complex 3D scaffolds that mimic the extracellular environment [26]. Various research groups have been researching scaffolds, among which porous scaffolds have been commonly used as vascular grafts, heart valves, and patches in recent years. The presence of pores enables cell infiltration, oxygen transport, and nutrient propagation into the tissue, as well as promotes drug transport and tissue angiogenesis [27,28,29,30,31]. It has also been reported that optimizing the size and shape of the pores reduces intimal hyperplasia and promotes angiogenesis [32,33,34]. Even so, promoting tissue regeneration for complete functional recovery in every tissue is still challenging.

This review will focus on tissue-engineered scaffolds applied in the cardiovascular field, namely, incredibly porous scaffolds. We primarily focused on the porous scaffolds applied in animal models or human clinical trials, including the most recent updates. Also, the results of using porous scaffolds, their status, and their challenges will be introduced.

## 2. Porous Scaffolds

Porosity is defined as voids in a material, further characterized by interconnections or throats between these voids, and media walls or struts that form a 3D structure (Figure 1). Traditionally divided into three subgroups by pore size—microporous above 50 nm, mesoporous between 2 and 50 nm, and microporous below 2 nm [35]. Materials with 1–100 nm pore sizes are described as nanoporous. Spherical, tubular, and random pore structures are commonly observed in porous materials, but new fabrication methods enable complex, high-resolution geometries [36,37] and topologies [38].

Considering the porosity of a material, its context is essential to assess the impact of porosity on its performance results. For tissue-engineered scaffolds, voids typically allow cell migration, oxygen, and nutrient influx to sustain these cultures and for metabolic waste outflow [28]. These voids are assumed to be saturated with interstitial fluid. Drug transport enables drug release by drug diffusion or material degradation [40,41]. Therefore, the penetration properties are strongly influenced by porosity, pore size, and interconnection. Capillary action is one of the penetration mechanisms which facilitates liquid uptake within the pores. As the pore or throat size decreases, size restrictions can limit the penetration of substances such as cells [20]. On the other hand, the smaller the pore size, the longer the fluid permeation distance due to capillary action. Thus, the effect of pore size on liquid penetration relies on the characteristics of the target scaffolds and the surface interactions. Porosity also affects the surface-area-to-volume ratio of the material. This is because as the percentage of porosity increases, the surface area of the material increases. Also, large pores with controlled porosity decrease the surface area of the material [41] The size of the wall or strut is also an essential factor in determining the surface-area-to-volume ratio. Materials with a high surface-area-to-volume ratio promote greater access for cell attachment [40] or facilitate drug release [41]. At the same time, the presence of pores also implies the possibility of an inflammatory response, a bacterial invasion, or platelet activation [42,43,44]. Increased porosity may also inversely affect their mechanical properties, such as scaffold strength or support. Although relatively influential, the specific choice of materials’ porosity also significantly impacts scaffold characteristics. In addition, particular compositions of materials can add some distinctive features, such as erodibility to change pore geometries in situ [45]. The pore size and porosity of porous scaffolds have also been reported to affect their biocompatibility. Their effects are summarized in Table 1.

## 3. Methods to Create Porous Scaffolds

There are several methods for creating porous scaffolds, which can be classified into subtractive and additive processes.

The subtractive method is a way to create pores in the scaffold whereby the scaffolds are formed with various substances in them, and then those substances are removed from the scaffold. Sphere templating, gas foaming, freeze-drying, and emulsion templating are included in this method. A polymer precursor is solidified in the sphere template with small spheres or salt particles as porogens [28,40]. In this method, it is possible to adjust the pores’ size by changing the particles’ size. Gas foaming methods use carbon dioxide as a porogen to create pores in the scaffolds instead of microparticles [28]. The advantage of this method is that it is biologically and environmentally friendly, as it does not use organic solvents. In the freeze-drying process, a colloidal suspension-combined polymer solution is frozen. Then, the ice particles are removed by melting. The pore sizes cannot be adjusted with this method [5]. Therefore, the above method does not allow the fabrication of a scaffold with a controllable porous microstructure. For these reasons, emulsion templating was produced. It is a way to produce monodisperse macroporous scaffolds, in which microfluidics allows liquid monodisperse foam and emulsion templates to be generated and then solidified [47] The limitation of this method is that the surfactant used for fabrication is challenging to remove. Therefore, the scaffolds have specific toxicity to cells [48].

Three-dimensional printing and electrospinning are known as additive methods. Due to recent improvements in 3D printing technology, it has been used more frequently as a porous scaffold creation method. In 3D printing, it is possible to create scaffolds using various materials such as polymers, metals, hydrogels, and living cells [49]. It is useful for creating larger pore scaffolds. For example, using hydrogel in a Bio-3D printer makes it possible to develop scaffolds with larger pores and arbitrary designs. However, high-precision equipment is expensive, and the application materials still need to be improved compared to other methods [48].

On the other hand, in the electrospinning method, the target polymer and volatile solvent are added to a positively charged syringe and injected from the needle tip toward the negatively charged collector. First, acceptable fibrous polymers are injected from syringes into the collector. Then, a scaffold is created by wrapping around the polymer fibers [50]. Unlike a 3D printer, it is excellent at creating scaffolds with high surface-to-volume. The limitation of electrospinning, however, is that the resulting pores are typically smaller than those in the scaffolds made with the subtracting methods [48].

## 4. Materials for Porous Scaffolds

Materials with porosity characteristics are characterized primarily by the pores’ size and the proportion or ratio of pores in the medium. For fibrous porous materials, the strut or fiber diameter is also essential. Pore and strut sizes are most commonly estimated using images. Methods such as optical microscopy, scanning electron microscopy (SEM), and transmission electron microscopy (TEM) can be used to see the internal structure [39].

Synthetic or natural polymers are mainly used in creating porous scaffolds in the cardiovascular area. Therefore, selecting biomaterials related to tissue formation is essential when creating platforms. Furthermore, since the tissue structure varies depending on the target tissue type, careful consideration is required in selecting the biomaterials.

Complex tissues, including tissue-engineered blood vessels or heart valves, are often created with synthetic porous polymers. In addition, fibrous proteins such as collagen and elastin constitute the extracellular matrix. On the other hand, fibrous proteins such as collagen are often used for a cardiac patch. Foreign natural polymers, which include chitosan and silk fibroin, are highly biocompatible. Therefore, these have been extensively studied as materials for single-use or multiple-use applications. Also, many reports have been made in recent years that composite materials that combine decellularized tissue with high tissue affinity and synthetic polymers have been developed. The following subsections are dedicated to different porous polymer types and their characteristics.

### 4.1. Synthetic Polymer

A balance between the tissue formation rate and the material degradation rate is one of the most critical factors for the tissue-engineered material to maintain its mechanical properties and proper tissue remodeling. The scaffold must withstand blood pressure during implantation and support proper neoplasia. Polylactic acid (PLA), polyglycolic acid (PGA), and copolymers are used in most of the previous studies [51,52,53]. These have also been used previously in absorbable sutures due to their FDA approval for human use. PGA is flexible and has very little inflammatory response. Also, it is durable enough to withstand mechanical stress equivalent to aortic pressure. PLA has similar structural and mechanical characteristics to PGA. However, PLA has a longer degradation time, which makes it possible to maintain suture tensile strength over a year [54]. However, PLA has a hydrophobic structure, which makes it unsuitable for use in the cardiovascular area [55]. Poly(L-lactide) (PLLA), polymerized with only L-lactic acid, lacks a hydrophobic structure. PLLA is one of the most studied polymers for cardiovascular tissue engineering applications [56]. Combining these materials shortens the decomposition time compared to using them alone. In addition, it is possible to modify the material’s mechanical properties with a different composition ratio. Because of its high biocompatibility and relatively slow degradation [57], PLLA is sometimes used with PGA.

Polycaprolactone (PCL) is another biodegradable polyester superior to PGA and PLA since it exhibits higher ultimate stress and tensile strength than native vessels. In addition, PCL has excellent biocompatibility and degrades slowly [58]. However, PCL polymer, which is also hydrophobic, is often used as a copolymer rather than as a single polymer to optimize cellular responses, thrombogenicity, and tissue invasion.

Polyurethane (PU) is often studied in materials combined with natural materials, because it is easy to control the mechanical characteristics and biodegradability by designing the structure of the crystalline domain based on the chemical formula [59,60].

### 4.2. Natural Polymer

Elastin and collagen have a role in mediating extracellular matrix (ECM) signals. In addition, they create elasticity and stiffness to maintain blood pressure and regulate smooth muscle cell function behavior. As collagen is the main component of the ECM, it promotes cell adhesion and proliferation. It also has a high biocompatibility with low antigenicity. Therefore, collagen has been investigated as a tissue-engineering material and is often used in cardiac patches [61,62].

Elastin is used to maintain elasticity with in vivo blood pressure since it has cell adhesion sites and helps to create bendability [63]. These characteristics contribute to the mechanical nature of arterial walls [64].

### 4.3. Silk, Fibroin, Chitosan

Biocompatible and biodegradable silk fibroin is a relatively frequently used biomaterial [65,66,67]. Silk protein is less immunogenic to humans and can promote angiogenesis and histogenesis. Furthermore, it can be used in several forms, including non-woven fabrics, sponges, or coatings, since it has decent thermal stability. For example, a bilayer vascular graft with sponge-coated and woven silk fibroin was used to evaluate a common carotid artery (CCA) bypass in a canine. The sponge layer of the graft was mainly degraded and replaced by fibrous tissue after one year [68].

On the other hand, chitosan is a relatively new material reported for use as a tissue-engineered vascular graft (TEVG). Thanks to its antibacterial effects and high hemostatic effect, chitosan has recently been studied more and more [69,70]. Chitosan has a low molecular weight and degree of deacetylation [71]. Chitosan fiber has been reported to degrade rapidly in vitro (5 days). It degrades even faster in vivo [72]. Therefore, chitosan is used as a combined graft with other materials to act as a quickly degraded material to facilitate cellular infiltration and vascular formation [73]. Using chitosan as a single material for cardiovascular tissue might be challenging since it degrades quickly.

### 4.4. Composite Material

Biodegradable polymers can be processed into porous bodies such as non-woven fabrics or sponge forms. It enables tailoring the design of target tissues with different cell permeabilities. Though the mechanical characteristics of biodegradable materials against pulsation and contraction are strong enough, they are not desirable for single-use cell attachment or proliferation. On the other hand, although natural materials have cell adhesion ability, they have poor mechanical properties. Nevertheless, endothelialization and mechanical characteristics must be emphasized [74]. For these reasons, hybrid scaffolds have emerged involving different materials and manufacturing, multi-layer construction, and molding into 3D. Another approach has been to develop bioactive substances such as collagen seeded with cells [75]. The seeded cells secrete ECM to reconstruct native tissue with permeability and biocompatibility [76]. However, the ideal balance of synthetic or biomaterials, including cells, to achieve adequate mechanical strength and characteristics remains challenging. In the following sections, we summarize the status of cardiovascular tissue engineering, including the latest research for each tissue.

## 5. Tissue-Engineered Vascular Graft

The targets of cardiovascular tissue engineering include blood vessels (arteries and veins), heart valves, and myocardium. Although they have similar characteristics in receiving systemic blood pressure, they have different histological features. Therefore, other materials are used for each tissue creation. In addition, the current state of research progress differs significantly with each tissue.

Identifying essential design requirements is critical to a successful TEVG [77]. First, the graft must have adequate burst pressure and compliance to persist against the blood flow in the graft. Second, the graft must be biodegradable in the host body, with minimal inflammation and immunogenicity to avoid rejection [78]. In addition, the graft should integrate into the native vessels in situ and create a vascular network inside the graft [79]. It means that the capacity for self-repair and remodeling is required to prevent graft failure. Important in small-diameter vessels, the inside lumen must be covered quickly with an endothelial layer after implantation to avoid thrombus formation [80]. Finally, grafting requires consideration of the material, cell type, and manufacturing process to generate the ideal TEVG to minimize adverse reactions [60,68,81,82,83] (Table 2).

### 5.1. Tissue-Engineered Arterial Graft (TEAG)

Matsuzaki et al. evaluated TEAG, a small-pore bilayer graft coated with an inner heparin-containing sponge layer and an electrospun outer layer in a sheep carotid artery. It emphasized the importance of scaling up from small animals such as mice to sheep. The speed of tissue regeneration differs in large animals; it is slower than that of small animals. Also, the decomposition speed of the scaffold has to slow accordingly [84]. Using slow-degrading PCL, the researchers fabricated a TEAG with a two-layer structure consisting of a PLCL sponge inner layer and an electrospun outer layer. In addition, they investigated the effect of the pore size after transplantation by fabricating different pore sizes of the outer layer [39]. The number of cells invading the sponge layer did not differ from the difference in pore diameter. However, the TEAG with a pore diameter of 4 μm showed more cell infiltration than a larger pore diameter of 7–15 μm (Figure 2). However, grafts with 4 μm pores could reduce regeneration of the smooth muscle layer in the inner layer more than grafts with larger pore sizes [85,86]. It is said that macrophage infiltration and tissue regeneration are promoted in scaffolds with pores of 30 μm or more. At the same time, grafts with 4 μm pores suppress macrophage infiltration and cytokine and matrix metalloproteinase secretion, activating the smooth muscle and secreting immature collagen. In this way, cell infiltration can be controlled by changing the materials used for TEAG and the pore sizes according to the tissue regeneration ability of the model animals and the conditions of the blood vessels to be transplanted. Also, the post-transplantation biological reaction and autologous tissue replacement can be adjustable. However, there are no reports of TEAG with longer-term follow-up of more than two years, and future reports are awaited.

### 5.2. Tissue-Engineered Venous Graft (TEVeG)

On the other hand, good mid-term results have been reported for TEVeG. Many studies reported good mid-term performance for up to 10 years [87,88]. Some groups achieved both absorbability and biocompatibility by creating a bilayer TEVeG. Unlike TEAG, TEVeG is not exposed to high arterial blood pressure. In addition, the pore size can be larger than TEAG to facilitate host cell infiltration.

Hibino et al. created grafts using an electrospun PGA outer layer on the inner layer of a PLCL sponge. As mentioned above, the sheep model slowly degrades the polymer material [89]. For the TEVeG with low internal pressure, an electrospun outer layer using PGA was adopted, since it decomposes relatively quickly. Sugiura et al. reported early results of their human clinical trials, which showed no aneurysm formation, graft rupture, infection, or calcification. Seven of the twenty-five patients developed intermediate stenosis during the course, and only one required balloon angioplasty [90]. After our design received FDA approval for the implant in patients in the United States, each of whom showed growth potential of the duct without severe comorbidities, including death. Even with clinical success, a high possibility of graft stenosis was observed again in six months. To clarify the mechanism of premature stenosis, Szafron et al. used a computational simulation model of our TEVeG with our previously collected data (Figure 3). It has been speculated that this narrowing, shown in clinical studies, is caused by an inflammatory mechanism and then spontaneously resolves [91].

Further investigation into this mechanism was performed by implanting the TEVeG into an ovine inferior vena cava interposition model, confirming the prediction of spontaneous resolution of TEVeG stenosis occurring in the mid-term period [92]. These results indicate that, although additional angioplasty could be avoided safely in patients with primary stenosis, adequate medical monitoring remains critical. Furthermore, the simulation also predicts that the mid-term stenosis can be reduced by modifying the scaffold design, such as the material content of the graft, to reduce the initial inflammation. The initial inflammation and consequent stenosis were suppressed by optimizing the TEVeG design, including the contents of the materials and structures.

The best results of clinical trials were reported for TEVeG, primarily in pediatric patients. Notwithstanding, there still needs to be more clinical trials. Current ongoing clinical trials are summarized from the ClinicalTrials.gov database in Table 3 [93]. Most of the problems are still in Phase 1 or 2, and with a limited number of cases. More results from these trials are awaited.

Regarding tissue-engineered vascular grafts, good long-term results have been reported with a combination of synthetic fibers. However, it still shows premature stenosis in TEVeG, and aneurysmal change or stenosis for TEAG. A large part of these issues may be solved by optimizing the content ratio of the materials.

## 6. Tissue-Engineered Heart Valve (TEHV)

Many research groups have attempted to create tissue-engineered heart valves, but the best design remains to be determined. In addition, it is essential to decide on the timing of the transplantation and to have the regenerative ability to replace it with a functional native valve over time. Although relatively good results have also been reported in vitro and preclinical studies, clinical applications require better functionality to replace autologous cells, which has yet to be realized [94].

TEHVs manufactured using porous scaffolds with bioabsorbable polymers have been extensively reviewed. This is because they can be naturally absorbed and metabolized by the native tissue [95]. In addition, synthetic polymers have the advantage of being reproducible, readily available, and tunable in size. Considering these advantages, many researchers have investigated the function of the remodeling potential in a TEHV, which was made of biodegradable polymers in large animal models [96,97,98,99,100,101] (Table 4).

Using a porous scaffold, a TEHV based on bioresorbable polymers can be seeded with autologous bone marrow mononuclear cells before transplantation, modulating the early inflammatory responses and remodeling cascades [102,103,104,105,106,107]. Also, the synthetic biodegradable polymer should be suitable for transcatheter applications to aortic or pulmonary valve replacement.

On the other hand, bis urea- and urethane-based supramolecular polymers are compatible with surgical [96,100] and transcatheter [99,106] implantation techniques. Furthermore, they have been used to manufacture resorbable valves and have shown good performance for up to 12 months [96,100]. Moreover, rapid cellularization, ECM deposition, and scaffold disassembly were observed in the scaffolds, confirming their remodeling potential. However, in a next-generation transcatheter aortic valve replacement (TAVR) [107], it was reported that the intra- and extra-valvular remodeling differed in the tissue remodeling [98,102,108]; further investigation is needed.

The Xeltis pulmonary valve conduit is made of a porous scaffold using an electrospun 2-ureido-4[1H]-pyrimidone (UPy) polymer; it is a heart valve designed to promote endogenous tissue regeneration slowly degrading to create a native heart valve [109]. The result was explored for 12 months in a sheep model. The scaffold was shown to degrade by macrophages. Then, it was replaced by smooth muscle cells and protein-rich deposits. After the replacement by native tissue, the valve had mild to moderate regurgitation. The conduit was covered with neointima in 2 months, and one of six animals had severe calcification deposits by 12 months. The Xeltis valve is currently in the clinical trial Xplore-1 study [108,110], evaluating the pulmonary valve conduits using biodegradable materials in 12 patients aged 2–12 years. A preliminary report of Xplore-1 performance at 24 months could have been more favorable. The echocardiographic evaluation showed the presence of moderate to severe pulmonary insufficiency in 11 of the 12 patients at six months after transplantation. In addition, an abnormal protrusion was observed at the valve. Xplore-2 is a second-generation device [110] that was introduced. The second-generation device had greater valve thickness at the commissure area since the 1st generation device revealed a commissural tear as the most typical failure mode. The second-generation device was implanted in six children 2–9 years of age. The preliminary reports from 12-month follow-ups showed good valve performance without dysfunction, but one case of valvular stenosis and another case required reoperation. Given the mixed results from these two trials, further clinical data with longer follow-ups are needed to determine the safety and efficacy of biodegradable polymer-based pulmonary valve conduits. Many TEHVs have been developed, some of which are hoped for in clinical trials, but the current results still necessitate improvements in the device to be used broadly [94].

Heart valves are subject to complex blood flow and pressure changes from pulsatile blood flow. To advance and improve tissue-engineered heart valve design techniques, it is essential to create novel designs to take over the current prosthetic valves. The research on TEHV may be further advanced by finding a rational selection and combination of materials for TEHV from the detailed degrading period and the order of degradation with a computational calculation model.

## 7. Cardiac Patch

Myocardial patches are required to have a function of continuous significant contraction. The sufficient density of cellular connections with the surrounding and non-cardiac myocytes is necessary to exert their physiological functions. It also requires extensive remodeling of the tissue architecture for rapid cell spreading, alignment, and replacement by cell-secreted ECM to the native scaffold, providing good biomechanics to allow continued tissue contraction. In addition, they are required to have specific characteristics. Therefore, developing suitable biodegradable biomaterials as candidates for cardiac tissue engineering and developing three-dimensional scaffolds with a particular shape, thickness, mechanical strength, and porosity to promote cell proliferation is essential.

Typical physical properties of the construct critical to this approach’s success include biocompatibility, the polymer’s chemical composition, protein-absorbing potential, surface energy, adhesion molecules, ability to nurture cells, controlled degradation rate, and permeation. The properties of the scaffold are the key, such as the porosity, suitable mechanical properties [111,112,113], ultrastructural properties [114], nano-topo graphics [115], electrophysiological stability, ability to facilitate vasculogenesis, possibility to create thick scaffolds of sufficient size, and strength [116,117]. Biomaterials are designed to mimic the complex native cardiac ECM, composed primarily of collagen [118]. Natural polymers (collagen monomers, gelatin), fibrin glues (fibrinogen), natural polysaccharides such as alginate/chitosan/hyaluronic acid, and synthetic polymers have been used as biomaterials (Table 5).

Sugiura et al. and Matsuzaki et al. used a PGA/PLCL woven fabric scaffold seeded with iPSC-derived cardiomyocyte or cardiac progenitor cells for myocardial regeneration. Their scaffold had 80% porosity and a 0.6 mm thickness. Their results showed that the structural strength was good with the ventricle replacement model, but the seeded cells were not found in the scaffold at 14 weeks. They reported that the seeded cells had paracrine effects to promote host myocardial regeneration [119,120]. As described in the TEAG section, creating a scaffold with large pores is still challenging, primarily in facilitating the invasion of human cardiomyocytes, since it has a size of more than 100 um. Also, some groups focus on creating the cardiac patch targeting paracrine effects. Czosseck et al. created a porous scaffold of PLGA and PLLA with various-sized pores. The largest pore is enabled to contain mesenchymal stem cells inside the scaffold. Then, the most prominent pores were sealed and secreted exosomes were released from the smaller pores, e.g., around 750 nm [121]. These methods to salvage dying cardiomyocytes are often reported with paracrine effects. However, it might not be a proper myocardial regeneration, since myocardial tissues need to be remodeled enough to regain their function as myocardial tissues.

**Table 5 bioengineering-10-00236-t005:** Studies of cardiac patches with porous scaffold.

Material	Animal Species	Number	Surgery or Intervention	Findings	Reference
PGA/PLCL with hiPS-CMs	Rat	6	RVOT reconstruction, with the cardiomyocyte seeded scaffold.	Seeded cells were not present in the patch after 4 weeks. The seeded cell might affect the host cardiac regeneration at 16 weeks.	[119]
PGA/PLCL with hiPS-CPCs	Rat	3	LV free wall reconstruction with CPC seeded scaffold.	Seeded cells disappeared at an early stage, no contribution to LV function, possibility of affecting angiogenesis at 9 months.	[120]
collagen+G-CSF	Rat AMI model	5 for each group	engrafting the collagen patch onto the injured myocardium	Effectively grafted, further increase in neovascularization with G-CSF	[122]
collagen with BMC+VEGF	Rat	3-4/group	RV free wall reconstruction with collagen patch.	Promoted cell proliferation within the graft, increased blood vessel density and reduced construct thinning.	[123]
Chitosan-hyaluronan/slik fibroin	Rat AMI model	11	epicardial placement on the injured area	Improved LV function, reduced LV dilation, also improved angiogenesis.	[124]

While natural materials such as collagen, fibrin, and synthetic polyglycolic acid have been extensively studied, Gaballa et al. created a 3-dimensional collagen type I scaffold with a solid porous foam. The collagen scaffold integrated into the myocardium and reduced the left ventricular dilatation. They also tested the effectiveness of the granulocyte colony-stimulating factor (G-CSF), which facilitates neovascular formation in the patch area [122]. Miyagi et al. also created a porous collagen patch. They tested the patch’s effectiveness combined with cell seedings as endothelial and bone marrow cells. They also added vascular endothelial growth factor (VEGF) to enhance the growth of the seeded cells. As a result, angiogenesis in the VEGF-added scaffold is facilitated, and cell survival and tissue formation are improved [123]. Also, several new compositions, such as silk fibroin and hyaluronic acid, an alginate/chitosan polyelectrolyte complex, have recently been introduced. For example, Chi et al. created a chitosan–hyaluronan/silk fibroin cardiac patch. They reported that this patch improved the left ventricular dilatation, wall thickness, and fractional shortening [124]. Interestingly, Yin et al. used a novel sponge-like conductive porous scaffold made of silk fibroin and polypyrrole (SP50). It has an electrical conductivity similar to the native myocardium when applied to cardiomyocytes on the scaffolds. In in vivo experiments, this conductive patch facilitated not only repairing the infarcted myocardium but restoring cardiac function. Furthermore, it promoted synchronous myocardium contractions in the scar area with the normal myocardium. This may be a promising regenerative method to prevent arrhythmia due to the irregular electrical activity in the scar area [125].

Various materials have also been tested for 3D gels and solid 3D porous sponges, including alginate, collagen, gelatin, polyglycolic, poly-lactic acid, and polyglycolic acid composites. A clear advantage of solid scaffolds over gels lies in the ease of manipulating arbitrary 3D forms and thicknesses over long periods. Stable scaffolds can also facilitate cell migration. Complex microtissues were also obtained with synthetic glycerol poly sebacate seeded with cardiac fibroblasts and neonatal cardiomyocytes. Beating cardiomyocytes in the microtissue have been reported, but proper contractility, which indicates proper cardiac function, has yet to be definitively demonstrated [126]. In myocardial regeneration, attempts have been made to control cell function by scaffolding using the electrospinning technique [117], and synthetic polymers such as PLGA and PCL are mainly used. Due to recent technological advances, methods of incorporating natural proteins, such as gelatin and collagen, into electrospun fiber have also been adopted. However, as previously mentioned in the TEAG section, a low-porosity structure is desirable to create a scaffold that can withstand intraventricular pressure. Still, it impedes cell penetration and efficiently incorporates ECM proteins required for cardiac regeneration.

On the other hand, some researchers have focused on developing macro-porous materials with hydrogels and have reviewed their potential applications in the field of tissue engineering [127,128,129,130,131]. Wang et al. reported the effectiveness of injectable hyaluronic acid hydrogels, which combined micro-RNA in the hydrogel [132]. This should be another merit of using hydrogels. They can connect not only cells but growth factors or even microRNAs.

Regarding the cardiac patch made by a porous scaffold, the number of studies using large animals is too limited to say it is promising. However, combining biomaterials such as scaffolds, ECM, hydrogels, cell components, and growth factors in a complex manner makes it possible to create a cardiac patch with good results by taking advantage of those materials.

## 8. Future Prospective

In this review, we reported the current state of tissue engineering using porous scaffolds in the cardiovascular field. Although all the fields are growing and developing remarkably, they still face challenges.

Good results have been reported in TEVG using grafts with porous scaffolds. However, in TEVG, graft stenosis due to remodeling in the mid-term stage remains a problem that should be improved. Creating a scaffold with an appropriate pore size will help solve the problem.

Regarding TEHV, there are still many aspects to be improved, such as the time course of replacement by host cells and its proper function as a valve. In addition, many attempts have been made to improve the polymer used, such as the leaflet structure. Promising results are awaited with a new combination of materials that will degrade but can create native valve tissue with proper function.

Cardiac patches are still mainly made from cells, but significant problems remain, such as a low survival rate of the cells during transplantation. In recent years, there have been increasing reports on porous scaffold-based patches with control-released growth factors that do not use cellular components. Still, the effects are limited with respect to salvaging cells. Therefore, creating a myocardial patch with regeneration potential is necessary by integrating technologies, such as porous scaffolds with microparticles containing cell components, synthetic polymers, and biological protein polymers.

If the engineering process involves cell seeding, it will take time and cost to be clinically approved. Research and development with off-the-shelf concepts, such as drug-eluting at the TEVG, are more likely to lead to clinical applications. In myocardial patch applications, it remains challenging to obtain sufficient myocardial regeneration without combining cell seeding. The discovery of factors that reacquire the multiplication ability in mature myocardium and adding such elements into scaffolds will enable adequate myocardial regeneration, even in off-the-shelf patches. It may lead the platforms to clinical applications more quickly.

## Figures and Tables

**Figure 1 bioengineering-10-00236-f001:**
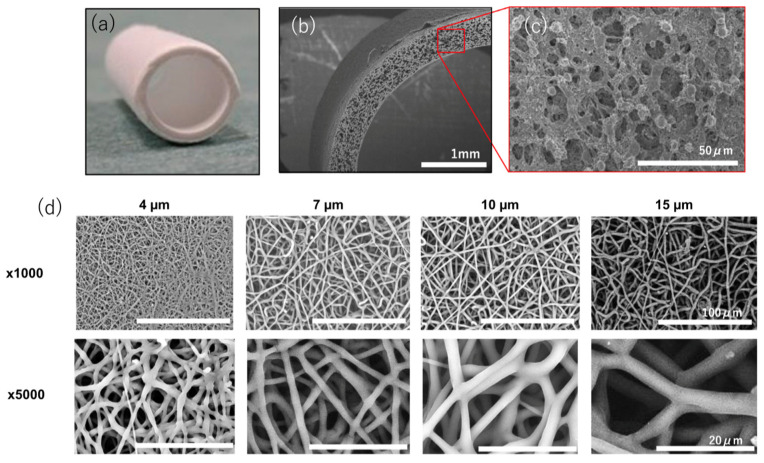
Porous scaffold. (**a**) Tissue-engineered arterial graft. (**b**,**c**) SEM pictures of the sponge-type porous scaffold. (**d**) SEM pictures of the electrospun scaffold with various pore sizes. Reprinted from ref. [39].

**Figure 2 bioengineering-10-00236-f002:**
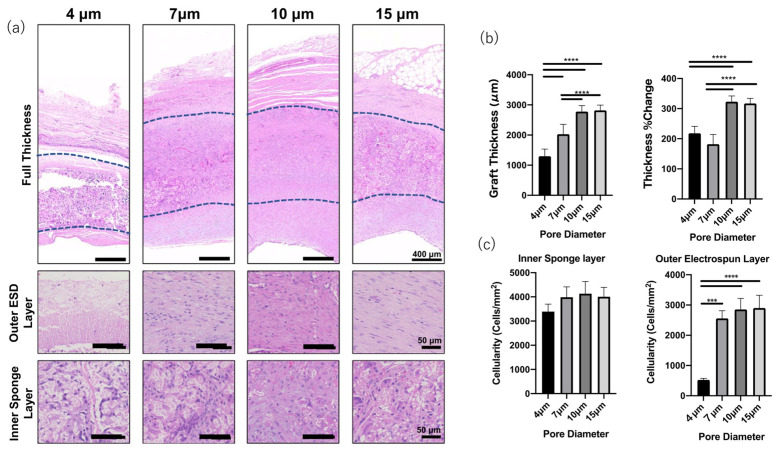
The difference in cell infiltration and wall thickness with different pore sizes. (**a**) H&E staining. (**b**) The wall thickness and cellularity difference depending on the pore size. **** *p* < 0.0001, *** *p* = 0.0001. (**c**) The cellularity difference depending on the pore diameter. Adapted from ref. [39].

**Figure 3 bioengineering-10-00236-f003:**
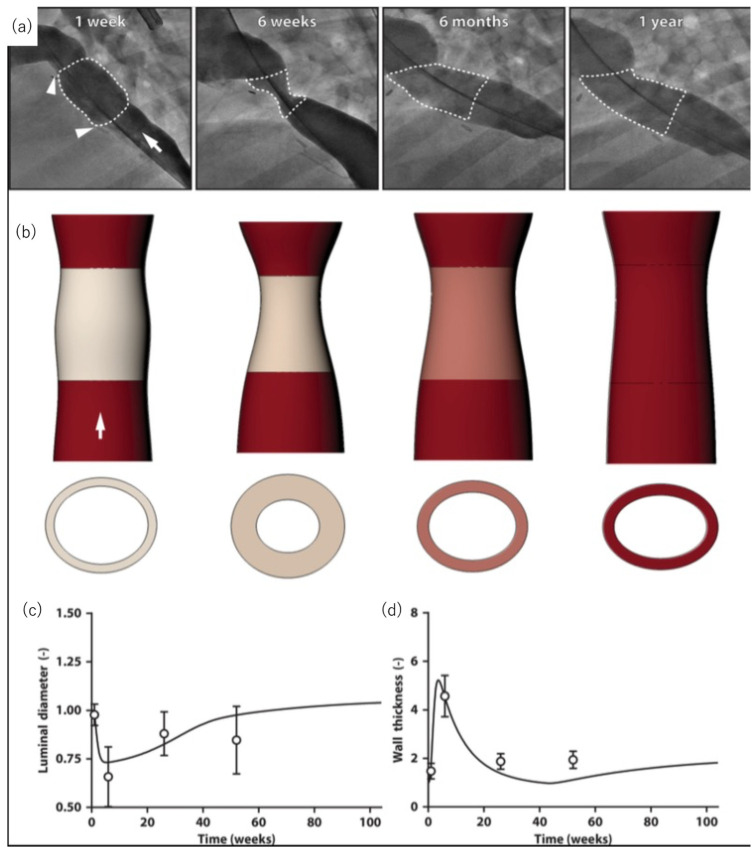
Spontaneous reversal of TEVG stenosis. (**a**) Angiographic images of TEVG with serial timepoint after the graft interposition. (**b**) Reconstructed images of serial IVUS and angiographic data, which shows the development of stenosis at six weeks, resolved spontaneously by six months after interposition. (**c**) Luminal diameter of mid-graft, (**d**) wall thickness measured by model simulations. Adapted with permission from ref. [92], Copyright 2020 The American Association for the Advancement of Science.

**Table 1 bioengineering-10-00236-t001:** The effect of pore size and porosity on biocompatibility. Adapted with permission from [46], Copyright 2023 John Wiley and Sons.

Response	Ideal Pore Size (um or ↑/↓)	Ideal Porosity (Value or ↑/↓)
Macrophage polarization	M1: <20 or >60 (at surface), 34 (intrapore) M2: 30–50 (at surface), >360	M1: ↓ M2: ↑
Angiogenesis	>5, ~40	↑
Hemocompatibility	<10 to limit platelet activation	↓to limit platelet activation; <50 mL H_2_O min^−1^ cm^2^ at 120 mmHg to limit leakage with anti-coagulants
Calcification	↑	↑; <5000 mL H_2_O min^−1^ cm^2^ at 120 mmHg to prevent inflammation
Reduction of fibrous capsule	30–40	↑

**Table 2 bioengineering-10-00236-t002:** Studies of TEVG with various materials of porous scaffolds.

Material	Animal Species	Number	Surgery or Intervention	Findings	Reference
PCL	rat	15	infra-renal abdominal aorta interposition with the graft	Rapid endothelialization, good patency and mechanical properties, insufficient regeneration of the vascular wall on the long term.	[81]
PU/PCL	rabbit	7	Carotid artery replacement.	Good anti-thrombosis, host cell infiltration, neotissue formation in 5 months.	[60]
PLA/PCL and PGA or PLLA	human	1	pulmonary artery recontruction	No evidence of graft occlusion or aneurysmal changes in 7 months.	[82]
Slik fibroin	dog	5	Carotid srtery replacement.	One of the implanted graft showed the pstency more than a year. Development of elastic fiber and reendothelialization.	[68]
PCL with decellularized Rat aorta	rat	6	infra-renal abdominal aorta interposition with the graft	Reduced neointimal hyperplasia. Progressed reendothelialization at 12 weeks.	[83]

**Table 3 bioengineering-10-00236-t003:** Current Clinical Trials of TEVG from the database of “ClinicalTrials.gov” [93].

Study Phase	Target Disease, or Situations	Scaffold	Original Estimated Enrollment	Outcome Measurement	Follow Up	Status
1	Single ventricle cardiac anatomy	synthetic polymer	4	Primary: Graft failure requiring intervention Second: Graft growth	3 years	completed
2	Vascular conduits for extracardiac total cavopulmonary connections	synthetic polymer	24	Primary: Safety and tolerability Secondary: Efficacy of TEVG determined by MRI	2 years	recruiting
1	Chronic venous insufficiency	ECM	15	Primary: Thrombosis, infection, surgical complications Secondary: symptoms of target disease, QOL, Graft durability, Flow abnormality, wall degeneration	1 year	recruiting
N/A	peripheral arterial disease	Collagen	20	Primary: Graft safety and adverse events Secondary: immunoreaction, graft patency, effect to symptoms a d anke-brachial index	2 years	Active, not recruiting
1	Hemodialysis access	Collagen	20	Primary: graft patency, intervention and adverse evemts Secondary: immunoreaction, patency and interventions	6 months	completed
N/A	Hemodialysis access	Collagen	40	Primary: Safety, tolerability and patency rate Secondary:	57 weeks	Active, not recruiting
N/A	Hemodialysis access	synthetic polymer	110	Primary: patency rate, freedom from device-related adverse events Secondary: implantation success rate, patency, interventions, infection	6 months	Recruiting
N/A	Hemodialysis access	synthetic polymer	20	Primary: device-related adverse events, patency Secondary: patency, adverse events	5 years	Active, not recruiting
N/A	Coronary artery bypass graft	synthetic polymer	15	Primary: Procedural success, device-related serious adverse events Secondary: intimal hyperplasia, patency, Major adverse events, mortality	5 years	Enrolling by invitation

**Table 4 bioengineering-10-00236-t004:** Studies of TEHV with various materials of porous scaffolds.

Material	Animal Species	Number	Surgery or Intervention	Findings	Reference
Upy-polyester-urethanes	Sheep	33	Transcatheter AVR	Good hemodynamics with acceptable degree of valve regurgitation	[99]
Upy-polyester-urethanes	Sheep	20	Surgical PVR	Durable hemodynamics, no stenosis or severe regurgitation	[97]
Upy-polyester-urethanes	Sheep	18	Surgical PVR	Neointima formation was observed, inflamation was peaked at 6 month while degradation peaked at 12 month.	[101]
Bisurea polycarbonate	Sheep	10	Surgical PVR	Remodeling with collagen and elastin synthesis, incomplete scaffold resorption in 12 months	[96]
Polycarbonate urethane urea and AZ31 magnesium alloy stent	Pig	5	Surgical PVR	Normal leaflet function in acute phase, no thrombosis or regurgitation	[100]
P4HB-gelatin	Sheep	4	Transcatheter PVR	Good hemodynamics, competence after implantation	[98]

## Data Availability

Not applicable.

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
