# Peer review of "The Application of Porous Scaffolds for Cardiovascular Tissues"

_bioengineering, 2023, doi:10.3390/bioengineering10020236_

Round 1

Reviewer 1 Report

The authors have drafted a review article named “The application of porous scaffold for cardiovascular tissue” and discussed the materials of scaffolds and their potential application and challenges. Corrections are needed before acceptance.

1.     Grammar mistakes should be addressed. For example, capitalize the letter at the beginning of the sentence, and don’t use capital if it is in the middle. Please pay attention to the consistency of the titles. Some periods are missing.

2. Section 2 explains other properties, such as electrical, mechanical, biosafety, and topography, in more detail.

3.     In Section 3, a discussion about the advantages and disadvantages of 3D printing and electrospinning should be included. I suggest adding other methods, such as emulsification and microfluidic.

4.     A brief summary of the clinical trials of scaffolds in Section 5 could be helpful.

5.     The authors have listed a bunch of applications in vitro, and a discussion of considering the commercialization of scaffolds should be included in the Future Perspective Section, like the current challenges, economic considerations, and future direction. 

Reviewer 2 Report

The paper entitled “The application of porous scaffold for cardiovascular tissue” is an in interesting domain but the review has some limitations.

1.      The paper needs to add the literature related to histology, anatomy of cardiovascular tissue and why the need of scaffolds for cardiovascular tissue engineering and what are the limitations in other therapeutics approaches, and the different kind of scaffolds utilized for cardiovascular tissue engineering.

2.      The introduction needs improvements.

3.      The authors need to add the literature regarding the different techniques utilized for the formation of porous scaffolds which isn’t mentioned in the current version.

4.      The author should have mentioned the significance and the novelty of this study in the literature as well and why this review is different from others as different study has been published in the same domain.

5.      There should be detailed literature regarding Porosity, Pore Size, and Interconnectivity of porous scaffold in cardiovascular tissue engineering. This should be added in form of a table or in figure.

6.      There are several typos errors throughout the manuscript, the final version needs to be revised.

Round 2

Reviewer 1 Report

The manuscript is accepted.